# High Diagnostic Accuracy of a Novel Lateral Flow Assay for the Point-of-Care Detection of SARS-CoV-2

**DOI:** 10.3390/biomedicines10071558

**Published:** 2022-06-30

**Authors:** Irene Giberti, Elisabetta Costa, Alexander Domnich, Valentina Ricucci, Vanessa De Pace, Giada Garzillo, Giulia Guarona, Giancarlo Icardi

**Affiliations:** 1Department of Health Sciences (DISSAL), University of Genoa, 16132 Genoa, Italy; irene.giberti25@gmail.com (I.G.); ec240583@gmail.com (E.C.); giadagarzillo@gmail.com (G.G.); icardi@unige.it (G.I.); 2Hygiene Unit, San Martino Policlinico Hospital-IRCCS for Oncology and Neurosciences, 16132 Genoa, Italy; valentina.ricucci@hsanmartino.it (V.R.); vanessa.depace@hsanmartino.it (V.D.P.); giuly.guarons@outlook.it (G.G.)

**Keywords:** SARS-CoV-2, COVID-19, lateral flow test, rapid antigen detection test, diagnostic accuracy, validation study

## Abstract

Highly accurate lateral flow immunochromatographic tests (LFTs) are an important public health tool to tackle the ongoing COVID-19 pandemic. The aim of this study was to assess the comparative diagnostic performance of the novel ND COVID-19 LFT under real-world conditions. A total of 400 nasopharyngeal swab specimens with a wide range of viral loads were tested in both reverse-transcription polymerase chain reaction and ND LFT. The overall sensitivity and specificity were 85% (95% CI: 76.7–90.7%) and 100% (95% CI: 98.7–100%), respectively. There was a clear association between the false-negative rate and sample viral load: the sensitivity parameters for specimens with cycle threshold values of <25 (>3.95 × 10^6^ copies/mL) and ≥30 (≤1.29 × 10^5^ copies/mL) were 100% and 50%, respectively. The performance was maximized in testing samples with viral loads ≥1.29 × 10^5^ copies/mL. These findings suggest that the ND LFT is sufficiently accurate and useful for mass population screening programs, especially in high-prevalence and resource-constrained settings or during periods when the epidemic curve is rising. Other public health implications were also discussed.

## 1. Introduction

Point-of-care (POC) diagnostics of SARS-CoV-2 enable a fast and decentralized testing model that should be part of the core global response to the ongoing COVID-19 pandemic [1]. Although traditional laboratory-based reverse-transcription polymerase chain reaction (RT-PCR) is currently considered the gold standard method for the diagnosis of SARS-CoV-2 infections, the use of POC tests has become increasingly common and tens of kits are commercially available [2,3]. Indeed, providing that the POC assays are sufficiently accurate, they may overcome some intrinsic limits of the laboratory-based RT-PCR, such as suboptimal turnaround times, and the availability of both sophisticated equipment and skilled personnel [4,5].

POC testing may be performed through both rapid molecular-based assays (e.g., all-in-one cartridge RT-PCR or reverse-transcription loop-mediated isothermal amplification) and various antigen-detecting rapid diagnostic tests (Ag-RDTs) that comprise (microfluidic) fluorescent and lateral flow immunochromatographic tests (LFTs) [6,7]. From the point of view of public health, these latter tests are particularly attractive for their relatively low cost, ease of use, and therefore, potential advantages in rolling out mass population testing programs [8].

LFTs are based on a specific antigen–antibody reaction with visual or automated readout achievable in a few minutes [9]. The available systematic evidence [10,11] suggests that the diagnostic accuracy of LFTs depends on a variety of factors, including brand, viral load, symptomaticity status, days since the onset of symptoms and sample storage conditions. It has been recently reported [10] that the pooled sensitivity and specificity of LFTs are 75.0% (95% CI: 71.0–78.0%) and 99.4% (95% CI: 99.3–99.4%), respectively. In this regard, skepticism pertaining to this suboptimal clinical performance is among the main determinants of primary care physicians’ attitudes towards the POC tests [12].

The International Federation for Clinical Chemistry and Laboratory Medicine (IFCC) recommends [13] that all laboratories should verify the analytical real-life performance of Ag-RDTs in at least 100 samples before their widespread routine use. Indeed, the sensitivity of Ag-RDTs declared by the manufacturers is usually over 95% [14], while numerous independent assessments performed under real-world conditions have shown significantly lower estimates [3,15,16]. For instance, a comparative evaluation of 122 CE-marked Ag-RDTs by Scheiblauer et al. [15] has demonstrated that the sensitivity of these assays may be as low as 0–30%. It is likely that this discrepancy is driven by the fact that the validation studies conceived by manufacturers for regulatory purposes may have low external validity as these evaluations are usually carried out on a limited number of well-characterized samples skewed towards high viral loads [17].

ND COVID-19 Ag test (NDFOS; Seoul, Republic of Korea) (henceforth referred to as “ND LFT”) has been recently developed for the qualitative detection of SARS-CoV-2 specific antigens in the human nasopharynx [18]. The objective of this study was to evaluate the real-world performance of the ND LFT in routinely processed nasopharyngeal (NP) specimens with a wide range of SARS-CoV-2 viral loads.

## 2. Materials and Methods

### 2.1. Compliance with Reporting Standards

The STARD (standards for reporting of diagnostic accuracy studies) statement [19] was adopted as a guideline for reporting (Appendix A, Appendix A).

### 2.2. Overall Study Design and Procedures

The study was conducted between 20 December 2021 and 30 April 2022 at the Liguria regional reference laboratory for COVID-19 diagnostics located at the San Martino Policlinico Hospital (Genoa, Italy). During this period, the overwhelming majority of samples tested in next-generation sequencing (NGS) belonged to the Omicron variant of concern (VOC).

For the main study, 400 NP specimens eluted in the universal transport medium (UTM) (Copan Italia; Brescia, Italy) with known RT-PCR results (100 positive and 300 negative) were consecutively collected. The sample size was pre-specified according to the Foundation for Innovative New Diagnostics (FIND) protocol [20], in which 100 positive and 300 negative samples are preferably recommended for clinical evaluation of Ag-RDTs. The index test was performed (see below) on fresh leftover and fully anonymized specimens and on the same day as RT-PCR.

To confirm the findings of the main study, a sample of 30 NP swab specimens positive for the pre-VOC strains with the D614G mutation (*n* = 10), Alpha (*n* = 10) and Delta (*n* = 10) VOCs were tested in the ND LFT. Beta and Gamma VOCs were rarely detected in Liguria and were, therefore, not tested. All these historically collected samples had a cycle threshold (Ct) <30 (which enables NGS) and were frozen at –80 °C. Of note, despite the storage at low temperatures, the viral concentration may have decreased over time [21], especially for the early collected pre-VOC strains stored for two years.

### 2.3. Index Test

ND LFT is intended as a POC assay for the rapid qualitative detection of SARS-CoV-2. The test is based on the principle of immunochromatography and detects both nucleocapsid and spike proteins. A clinical evaluation study on 84 NP swab samples disclosed on the product leaflet reported a sensitivity and specificity of 94.1% (95% CI: 80.3–99.3%) and 100% (95% CI: 92.9–100%), respectively.

For the index test, a 50 µL aliquot of each sample was absorbed on the sterile cotton flock provided with the ND LFT [15]. The soaked swabs were then eluted in the test buffer and stirred for 15 s, as per the manufacturer’s instructions. Finally, three drops of the treated processing solution were put into the sample card and results were assessed visually after 15 min by two researchers; any eventual disagreement was solved by a third researcher. Briefly, the coloration of both test and control lines stands for a positive result, while the test was dubbed “negative” when only the control line was colored. The test instead is invalid when no coloration of the control line occurred, independently of the aspect of the test line.

### 2.4. Reference Test

The reference test was an extraction-free multiplex RT-PCR adopted by the San Martino Policlinico Hospital as a standard-of-care assay for the molecular diagnosis of SARS-CoV-2. The protocol for this method has been previously validated [22], showing a perfect agreement with a standard extraction-based RT-PCR technique. Briefly, 5 µL of fresh (<8 h) NP swab specimens were diluted with molecular-grade water in a 1:3 ratio and set up directly for RT-PCR. This latter was run on the CFX96 thermal cycler (Bio-Rad Laboratories, Hercules, CA, USA) by using the Allplex 2019-nCoV multiplex kit (Seegene; Seoul, Republic of Korea). The Allplex 2019-nCoV assay detects nucleoprotein (N), RNA-dependent RNA-polymerase (RdRp) and envelope (E) gene regions. For each reaction, 5 µL of the previously obtained mixture in a final volume of 20 µL was used. The following thermal profile was used for amplification: 50 °C for 20 min, 95 °C for 15 min, 45 cycles at 95 °C for 10 s, 60 °C for 15 s with the first acquisition and 72 °C for 10 s with the second acquisition on the CFX96 instrument. Amplicons were tested by FAM (E), HEX (internal control), Cal Red 610 (RdRP) and Quasar 670 (N) fluorophores and the resulting amplification curves were read by means of the 2019-nCoV viewer (Seegene; Seoul, Republic of Korea) according to the manufacturer’s instructions. Specimens showing Ct values <37 for at least two genes were considered positive. Positive samples were further categorized into very high (Ct < 25), high (Ct 25–29.9) and moderate (≥30) viral loads [15,23].

SARS-CoV-2 concentration (copies/mL) was determined by linear extrapolation of the Ct values plotted against the known viral concentration. For this purpose, the AccuPlex SARS-CoV-2 reference kit (SeraCare; Milford, MA, USA) with a concentration of 5000 copies/mL was used. The resulting extrapolation for the Ct cut-offs of 25 and 30 was approximately 3.95 × 10^6^ and 1.29 × 10^5^ copies/mL, respectively. The obtained regression coefficients were in line with a previous linear interpolation between the viral concentration and Ct values displayed by the Allplex 2019-nCoV assay [24].

### 2.5. Statistical Analysis

For descriptive purposes, categorial and approximately normally distributed continuous variables were expressed as percentages with 95% confidence intervals (CIs) and means with standard deviations (SDs), respectively. Concordant, true positive and true negative rates were reported as the overall agreement, sensitivity and specificity parameters (with 95% CI), respectively. These latter were computed overall and by Ct category. An optimal Ct value cut-off that maximizes both sensitivity and specificity of the index test was estimated using Youden’s J statistic.

Data analysis was performed in OpenEpi v. 3.01 [25] and R stat packages v. 4.1.0 (R Core Team; Vienna, Austria) [26].

## 3. Results

A total of 100 positive and 300 negative samples were tested in both RT-PCR and ND LFT. There were no invalid (no coloration of the control line) ND LFT results. Among RT-PCR positive samples, the average Ct values were 24.4 ± 4.3, 26.5 ± 4.2 and 24.3 ± 4.4 for genes E, RdRp and N, respectively.

Of 400 ND tests performed, 3.75% (*n* = 15) of samples showed discordant results and all of these were judged false-negatives (Appendix A). As shown in Table 1, the overall sensitivity and specificity of the ND LFT were 85.0% and 100%, respectively. All false-negative results had relatively low viral loads (Figure 1). Indeed, while the sensitivity of the ND LFT was 100% for samples with Ct values < 25, it fell down to 72.0% and 50.0% for samples showing Ct values of 25–29.9 and ≥30, respectively (Table 2). According to Youden’s J, the optimal Ct threshold value was 28, which roughly corresponds to the viral concentration of 5.07 × 10^5^ copies/mL.

Finally, when 30 positive frozen samples were tested, the overall positive agreement was 80.0% (95% CI: 62.7–90.5%). In particular, six (60%), eight (80%) and ten (100%) samples positive for the D614G strains, Alpha and Delta VOCs turned out positive in the ND LFT.

## 4. Discussion

In this real-world study, a relatively large number of NP samples were tested in the novel ND LFT. The overall sensitivity and specificity of the test were 85.0% and 100%, respectively. While our specificity estimate was in line with that declared by the manufacturer, the sensitivity was about 9% lower than that reported on the package leaflet (94.1%) and outside the observed 95% CI of 76.7–90.7%. On the other hand, despite a relatively wide 95% CI for the sensitivity parameter, the ND LFT met the internationally recognized performance criteria. Indeed, according to the World Health Organization (WHO) [27] and European Centre for Disease Prevention and Control (ECDC) [23], Ag-RDTs should meet minimum performance requirements of ≥80% sensitivity and ≥97% specificity. However, it should be borne in mind that the overall sensitivity depended on the relative distribution of the true positive samples according to their Ct values. For instance, studies skewed towards including more specimens with very high or high viral loads will likely produce more favorable outcomes and vice versa. For instance, if in our study samples with very high, high and moderate viral loads were equally distributed, the overall sensitivity would drop to 74% (results not shown). It is indeed likely that the high level of statistical heterogeneity reported in the available meta-analyses [3,10,11] is also driven by the distribution of Ct values. Future large-scale studies with more true positive samples would be beneficial for the post-marketing life cycle evaluation of the ND LFT.

As we mentioned above, the performance of the ND LFT was made worse with decreasing viral loads: while the sensitivity was 100% for samples with Ct values <25, it dropped to 50% for specimens showing Ct ≥ 30. This finding is in line with numerous [3,10,11,16,28,29,30] pooled analyses conducted so far. According to the Youden’s index, the best trade-off Ct value that maximized sensitivity of the ND LFT was 28, which corresponds in our study to the viral concentration of approximately 5 × 10^5^ copies/mL. In this regard, it has been demonstrated [31] that at the population level, low median Ct values with a weak negative skewness may prove that the SARS-CoV-2 incidence is growing. In contrast, when the median Ct values are high and their distribution is strongly skewed to the left it may testify to the epidemic’s decline. Our data, therefore, suggest that LFTs and other Ag-RDTs with similar diagnostic accuracy parameters may be particularly useful for mass screening programs when the number of new cases is growing.

Our results corroborate the recommendations on the implementation of Ag-RDT testing programs issued by the ECDC [23]. They suggest that Ag-RDTs should be used for testing symptomatic individuals; concerning the screening of asymptomatic persons, Ag-RDTs should be used when the positivity prevalence is high (≥10%). Indeed, at low prevalence, the expected positive predicted value would be low. In several instances, negative Ag-RDT results should be confirmed by another method, such as RT-PCR [23]. On the other hand, despite RT-PCR being currently considered the gold standard technique [2,3], up to 58% of COVID-19 patients may initially test negative in RT-PCR and the impact of misdiagnosis enlarges with the increasing incidence [32]. In this regard, the so-called “test, re-test, re-test” strategy [33] of implementing Ag-RDTs could be considered, especially in resource-constrained settings. This strategy is based on the multiplication rule for independent events and may drastically reduce the probability of false-negative results. For example, the probability of a false-negative result for the second and third LFT test with a known sensitivity of 50% will drop to 12.5% and 6.25%, respectively. Three LFT tests may be both less expensive and faster than one RT-PCR run [33].

The most important limitation of this study is that the validation was conducted on anonymized leftover samples, and therefore, no clinical data (e.g., symptomaticity status and days passed between the onset of symptoms and the swab) were available. The stratification of positive samples by RT-PCR Ct values may partially address this shortcoming since the patient’s viral load may be considered a proxy of disease severity and evolution. For instance, symptomatic individuals tend to have higher average viral loads than those who are asymptomatic [34]. Analogously, the maximal viral load typically arises early during the disease with a subsequent exponential decay leading to the viral clearance [35]. Another shortcoming is that in the main study only samples positive for the Omicron VOC were tested, and therefore, the reported results may be not generalizable to diverse SARS-CoV-2 populations. To partially address this limitation, we evaluated some samples positive for earlier circulating strains. However, this feasibility assessment performed on a limited number (*n* = 30) of frozen samples was not sufficiently powered to establish diagnostic accuracy. In summary, future large-scale studies conducted on patients with different clinical presentations and affected by novel SARS-CoV-2 variants are warranted.

In conclusion, in our study, the ND LFT showed an acceptable diagnostic accuracy profile for detecting SARS-CoV-2 in NP samples with high viral loads (Ct < 30). The test performance suggests its usefulness in mass population screening programs, especially in situations when the epidemic curve is rising and in high-prevalence settings.

## Figures and Tables

**Figure 1 biomedicines-10-01558-f001:**
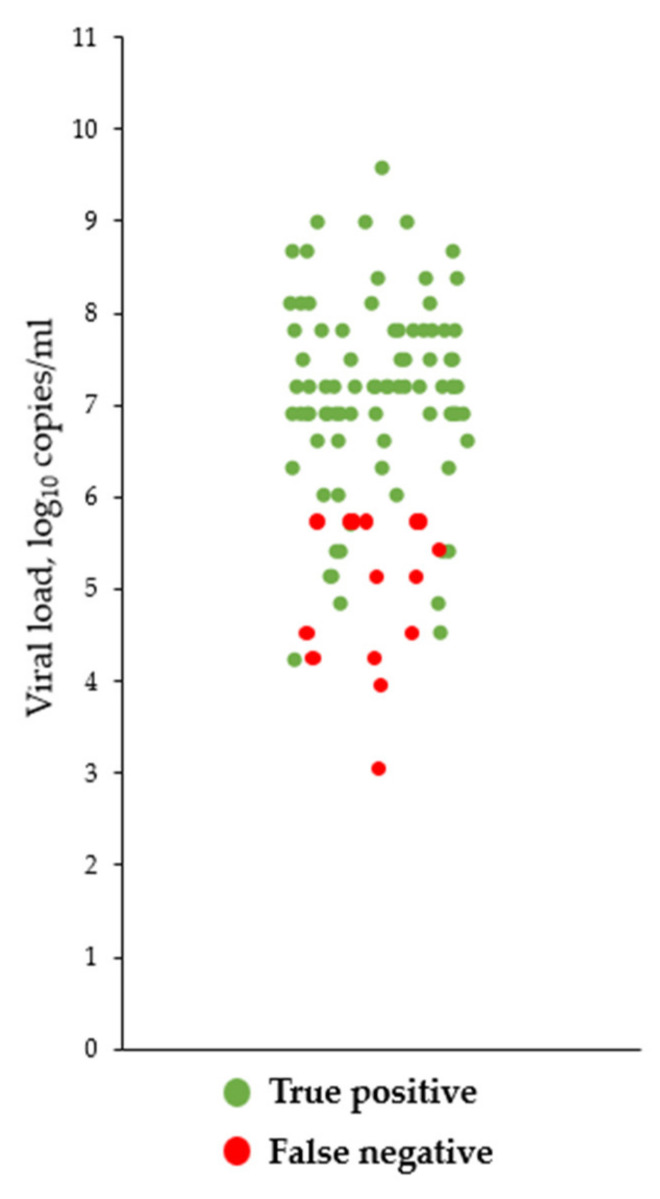
Distribution of true positive and false-negative samples, by viral load.

**Table 1 biomedicines-10-01558-t001:** Diagnostic accuracy parameters of the ND lateral flow test.

Accuracy Parameter	Estimate, %	95% CI
Overall accuracy	96.3	93.9–97.1
Sensitivity	85.0	76.7–90.7
Specificity	100	98.7–100

**Table 2 biomedicines-10-01558-t002:** Sensitivity of the ND lateral flow test, by cycle threshold category.

Cycle Threshold (*n*)	Estimate, %	95% CI
<25 (59)	100	93.9–100
25–29.9 (25)	72.0	52.4–85.7
≥30 (16)	50.0	28.0–72.0

## Data Availability

All relevant data are within the manuscript. Further details may be obtained from the corresponding author upon a reasonable request and prior permission of the study funder.

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
