# Peer review of "High Diagnostic Accuracy of a Novel Lateral Flow Assay for the Point-of-Care Detection of SARS-CoV-2"

_biomedicines, 2022, doi:10.3390/biomedicines10071558_

Round 1
Reviewer 1 Report
The authors wanted to assess the comparative diagnostic performance of the new ND Covid-19 Lateral Flow Immunochromatographic tests (LFT) in real-life conditions with routinely processed nasopharyngeal specimens , with a large range of SARS-Cov2 viral loads. 400 nasopharyngeal specimens with a large range of viral loads were tested using both reverse-transcription polymerase chain test and ND LFT. The overall sensitivity and specificity were 85% with a large CI from 76.7 to 90.7% and 100% respectively. The authors found also a strong association between the false negative rate and viral loads whose the sensitivity parameters for cycle threshold values of <25 and >=30 were 100 and 50% respectively. In conclusion, the ND LFT is sufficiently accurate and efficacious for mass population screening programs mainly during the Covid-19 waves.
ANALYSIS OF THE ARTICLE
INTRODUCTION: the history of the diagnostic tests and the interest of the LFT are well reported. The factors that can weaken the accuracy of the LFT tests are well described.
METHODS: the study design is well conducted , in a single center allowing a standardized and an homogenous practice. A cohort of 400 specimens with 100 positive and 300 negative cases, were diagnosed by the known RT- PCR tests. The biological tests were performed with a strong quality control. For the confusing findings of the main study, a sample of 30 positive specimens were tested with the ND LFT. Statistical analysis is well conducted.
RESULTS: They are clearly described with details. From a total of 100 positive and 300 negative samples, there were no invalid ND LFT results. Only 15 samples showed discordant results and they were judged as false -negative. All false-negative specimens had low viral loads. The ND LFT specimens had a sensitivity by 100% for samples with CT values < 25 and it fell to 72.0% and 50 % for samples with CT values by 25-29.9 and >=30 respectively. Figure 1 is very useful.
DISCUSSION: it is built around the efficiency and the reliability of the new process. The results satisfy the international recognized performance criteria with a minimum performance level by 80% for the sensitivity and >= 97% for the specificity, despite a very large diveristy of viral loads equally distributed. The authors recall the limit of the tool when the sensitivity value decreases with the decrease of the viral load, in line with several prior analyses. The authors close their discussion recalling the good use of the tool
CONCLUSIONS: the study shows an acceptable diagnostic accuracy profile to detect SARS-Co2 from nasopharyngeal samples with high viral loads. They suggest its usefulness in mass population screening
REMARKS FOR THE AUTHORS: to improve the article , we suggest to:
1) discuss the limit of the wide Confidence Interval of the sensitivity value between 76.7 AND 90.7 %. and the problem of the false-positive results. The measure of the Positive Predictive Value could be interesting
2) explain the interest to test 30 nasopharyngeal specimens with the ND LFT to verify the confusing findings
Author Response
Comment: The authors wanted to assess the comparative diagnostic performance of the new ND Covid-19 Lateral Flow Immunochromatographic tests (LFT) in real-life conditions with routinely processed nasopharyngeal specimens, with a large range of SARS-Cov2 viral loads. 400 nasopharyngeal specimens with a large range of viral loads were tested using both reverse-transcription polymerase chain test and ND LFT. The overall sensitivity and specificity were 85% with a large CI from 76.7 to 90.7% and 100% respectively. The authors found also a strong association between the false negative rate and viral loads whose the sensitivity parameters for cycle threshold values of <25 and >=30 were 100 and 50% respectively. In conclusion, the ND LFT is sufficiently accurate and efficacious for mass population screening programs mainly during the Covid-19 waves.
INTRODUCTION: the history of the diagnostic tests and the interest of the LFT are well reported. The factors that can weaken the accuracy of the LFT tests are well described.
METHODS: the study design is well conducted, in a single center allowing a standardized and an homogenous practice. A cohort of 400 specimens with 100 positive and 300 negative cases, were diagnosed by the known RT- PCR tests. The biological tests were performed with a strong quality control. For the confusing findings of the main study, a sample of 30 positive specimens were tested with the ND LFT. Statistical analysis is well conducted.
RESULTS: They are clearly described with details. From a total of 100 positive and 300 negative samples, there were no invalid ND LFT results. Only 15 samples showed discordant results and they were judged as false -negative. All false-negative specimens had low viral loads. The ND LFT specimens had a sensitivity by 100% for samples with CT values < 25 and it fell to 72.0% and 50 % for samples with CT values by 25-29.9 and >=30 respectively. Figure 1 is very useful.
DISCUSSION: it is built around the efficiency and the reliability of the new process. The results satisfy the international recognized performance criteria with a minimum performance level by 80% for the sensitivity and >= 97% for the specificity, despite a very large diveristy of viral loads equally distributed. The authors recall the limit of the tool when the sensitivity value decreases with the decrease of the viral load, in line with several prior analyses. The authors close their discussion recalling the good use of the tool
CONCLUSIONS: the study shows an acceptable diagnostic accuracy profile to detect SARS-Co2 from nasopharyngeal samples with high viral loads. They suggest its usefulness in mass population screening
Reply: Thank you for your interest in our paper. No changes required.
Comment: REMARKS FOR THE AUTHORS: to improve the article, we suggest to:
1) discuss the limit of the wide Confidence Interval of the sensitivity value between 76.7 AND 90.7 % and the problem of the false-positive results. The measure of the Positive Predictive Value could be interesting
Reply: As suggested, we have now discussed on the relatively wide 95% CI for the sensitivity parameter. Considering that the observed specificity was 100%, the expected positive predictive value is 100%, independently from the prevalence of SARS-CoV-2 positivity.
Comment: 2) explain the interest to test 30 nasopharyngeal specimens with the ND LFT to verify the confusing findings.
Reply: Thank you for this comment. Given that the main study with 400 samples was conducted during the period when only the Omicron variant of concern was circulating, an additional assessment on 30 samples positive for the earlier circulating variants was also conducted. As required, we have now discussed on this approach and its limitations.
Reviewer 2 Report
The present study is original and important. Its practical value is significant and the content is convincingly proving it.
My major comments and suggestions are as follows:
1. Please, read carefully the manuscript and make grammatical corrections;
2. Please, explain in more details the output of the statistical treatment - meaning and significance of sensitivity, specificity and overall accuracy values
Author Response
Comment: The present study is original and important. Its practical value is significant and the content is convincingly proving it. My major comments and suggestions are as follows:
Reply: Thank you for your interest in our paper. All your comments have been addressed.
Comment: 1. Please, read carefully the manuscript and make grammatical corrections;
Reply: The whole manuscript has been now revised.
Comment: 2. Please, explain in more details the output of the statistical treatment - meaning and significance of sensitivity, specificity and overall accuracy values.
Reply: This has been done.
Reviewer 3 Report
This is a validation study to demonstrate the accuracy of a newly developed lateral flow assay for detecting SARS-CoV-2 with the anonymized specimen. The results are good for the use of the clinical practice. The description and presentation of the study are appropriate to demonstrate. As the authors mentioned, a major limitation is that no clinical information on the specimen was available. A future study is warranted to show the association between clinical characteristics and this lateral flow assay results.
Figure S1 and Table S1 should be published as supplemental materials, not 'Non-published Material.'
Author Response
Comment: This is a validation study to demonstrate the accuracy of a newly developed lateral flow assay for detecting SARS-CoV-2 with the anonymized specimen. The results are good for the use of the clinical practice. The description and presentation of the study are appropriate to demonstrate. As the authors mentioned, a major limitation is that no clinical information on the specimen was available. A future study is warranted to show the association between clinical characteristics and this lateral flow assay results.
Reply: Thank you for your interest in our paper. We have now mentioned in the discussion that future studies should also consider clinical characteristics of patients.
Comment: Figure S1 and Table S1 should be published as supplemental materials, not 'Non-published Material.'
Reply: Both Figure S1 and Table S1 are within the Supplementary Material.